# Performance Evaluation of Particulate Matter and Indoor Microclimate Monitors in University Classrooms under COVID-19 Restrictions

**DOI:** 10.3390/ijerph18147363

**Published:** 2021-07-09

**Authors:** Laurentiu Predescu, Daniel Dunea

**Affiliations:** 1Department of Food Engineering, Faculty of Environmental Engineering and Food Science, Valahia University of Targoviste, Aleea Sinaia No. 13, 130004 Targoviste, Romania; laurentiu.predescu@valahia.ro; 2Department of Environmental Engineering, Faculty of Environmental Engineering and Food Science, Valahia University of Targoviste, Aleea Sinaia No. 13, 130004 Targoviste, Romania

**Keywords:** PM2.5, PM1, size segregated mass fractions, thermal microclimate, predicted mean vote (PMV), predicted percentage of dissatisfied (PPD), particle counter, mask wearing, acute dose rate (ADR)

## Abstract

Optical monitors have proven their versatility into the studies of air quality in the workplace and indoor environments. The current study aimed to perform a screening of the indoor environment regarding the presence of various fractions of particulate matter (PM) and the specific thermal microclimate in a classroom occupied with students in March 2019 (before COVID-19 pandemic) and in March 2021 (during pandemic) at Valahia University Campus, Targoviste, Romania. The objectives were to assess the potential exposure of students and academic personnel to PM and to observe the performances of various sensors and monitors (particle counter, PM monitors, and indoor microclimate sensors). PM1 ranged between 29 and 41 μg m^−3^ and PM10 ranged between 30 and 42 μg m^−3^. It was observed that the particles belonged mostly to fine and submicrometric fractions in acceptable thermal environments according to the PPD and PMV indices. The particle counter recorded preponderantly 0.3, 0.5, and 1.0 micron categories. The average acute dose rate was estimated as 6.58 × 10^−4^ mg/kg-day (CV = 14.3%) for the 20–40 years range. Wearing masks may influence the indoor microclimate and PM levels but additional experiments should be performed at a finer scale.

## 1. Introduction

Particulate matter (PM) represents small particles that range in size from less than 1 micron to about 100 microns [1]. These particles remain suspended in the environment for a long time. When they are inhaled in various microenvironments, they penetrate deep into the lungs, having the direct effect of increasing morbidity in the exposed individuals [2]. These particles are also responsible for significant economic losses due to their corrosive properties and adhesion to the surfaces on which they eventually settle down [3].

A particle diameter of fewer than 10 microns was considered to protect human health because it can enter the thoracic cavity through respiration. Inhalable particles larger than 10 microns in diameter have a much lower potential effect on health. In the United States, this standard entered into force on 31 July 1987 [4]. Under the Air Quality Directive 2008/EC/50, the European Union has set limit values for outdoor particulate matter (PM10 and PM2.5) regarding the protection of human health, (https://eur-lex.europa.eu/eli/dir/2008/50/oj, accessed on 5 June 2021). 

The main adverse effects on the health of the population are generated by particles with an aerodynamic diameter of fewer than 10 micrometers, which pass through the nose and throat, reaching the lung alveoli causing inflammation and intoxication [5]. The most affected vulnerable groups are people with cardiovascular and respiratory diseases, children, the elders, and asthmatics [6]. Long-term exposure to a constant concentration of suspended particles can cause cancer and premature death [7,8].

Young people inhale a higher amount of air, and depending on its load, it results in a more pronounced exposure to the concentrations of air pollutants [9]. It has been demonstrated that PM aggravates the symptoms of asthma, manifested by breathing difficulties, cough, chest pain, etc. [10]. More attention should be given for the fractions lower than 4 µm considered as a threshold for the respirable fraction [11]. 

PM is a potential carrier of allergens or microorganisms, including viruses, which are unable to survive separately. Studies have found that PM1 and PM2.5 are virus-carrying particles that can be inhaled into the lower respiratory tract triggering an immune response and increasing secretions and expressions of inflammatory cytokines [12]. Recent epidemiological studies pointed out that the smaller the particle diameter, the higher the possibility of deeper penetration into the lungs up to the alveolar level [13,14]. A study in Belgium involving measurements of a large number of indoor and outdoor air pollutants at 30 elementary schools (90 classrooms total) demonstrated that concentrations of many chemicals including various fractions of PM were much higher indoors than outdoors. It also showed high variability in concentrations between classrooms [15]. A recent study found that human walking induced indoor PM2.5 resuspension leads to an increase in indoor particulate matter requiring regular cleaning of the indoor dust to reduce the secondary pollution caused by indoor activities [16]. Within the university indoor environments, the main source of a PM is related to the resuspension of particles because of mobility of occupants and the outdoor concentrations enter indoors and remain inside the closed environment [17]. In these microenvironments, the students and academic personnel spend several hours being exposed to the concentrations of various air pollutants including PM. Thus, it is important to know the levels of exposure by using reliable sensors, methodology, and modeling algorithms [18].

The reference methods for measuring the mass concentration of suspended particulate matter (e.g., EN 12341:2014) are not capable of producing real-time data. EU regulations allow the use of equivalent methods if the equivalence of methods complies with the standards. It establishes a procedure for quantifying the correspondence between reference methods and equivalent methods through a series of parallel field measurements (CEN EN 16450:2017) [19].

The aim is for equivalent instruments to provide daily data with a lower measurement uncertainty than required by the ambient air quality directive (±25% with a 95% confidence level) at concentrations close to the limit value. Gravimetric samplers with filters are mainly based on operators for their maintenance and operation, especially if the filters are changed daily and the subsequent weighing of the filters is necessary. Quality assurance of measurements must be strictly applied, due to the additional ways in which errors occur (e.g., handling of samples, transport, storage, and weighing). In addition, there may be delays (sometimes several days) between the sampling stage and the reporting stage for the filters to be weighed in a laboratory with special air conditioning conditions. This is detrimental to the updating of information for the protection of public health and is contrary to the reporting requirements of the directives in force [20]. Furthermore, the utilization of gravimetric samplers in indoor studies is difficult because of the information is required during the occupation of the analyzed room, which assumes various time intervals. 

To overcome these shortcomings of the reference method, numerous PM10 and PM2.5 particle-measuring instruments (TEOM analyzer, beta attenuation monitor, optical devices, etc.) and later on, a series of low cost sensors have been developed [21]. Some of the existent equipment can also be used to measure the submicrometric fraction by changing the impactor or cyclone (particle pre-selection devices according to the particle diameter) or use a cascade impactor for several fractions [22].

Particle analyzers using optical methods are based on estimating the interaction between suspended particles, on the one hand, and visible or infrared spectrum radiation or a laser beam, on the other. Nephelometric systems that operate based on a short closed path to propagate the emitted radiation measure the light scattering responsible for most of the total amount of light extinction. The advantage of these instruments is that a single device can simultaneously monitor the fine particles—PM10 and PM2.5, but also the submicrometric ones (PM1) [23]. Among other applications in which optical monitors have proven their versatility are the studies of air quality in the workplace and indoor environments [24].

The aim of the study was to perform a screening of the indoor environment regarding the presence of various fractions of particulate matter (PM) and specific thermal microclimate in a classroom occupied with students in March 2019 (before COVID pandemic) and in March 2021 (during pandemic) at Valahia University Campus, Targoviste, Romania. The research hypothesis presumed differences between the groups of students on PM load and indoor microclimate in the university classroom due to the number of occupants and the use of masks. The objectives were to further assess the potential exposure of students and academic personnel to PM and to observe the performances of various sensors and monitors (particle counter, PM monitors, and indoor microclimate sensors). The results are expected to provide contextual data for environmental exposure assessment and useful insights about the influence of wearing masks on the indoor microclimate modifications, and give perspectives for the future of sensors that can support medical and occupational health and safety research in indoor environments. Furthermore, the study provides several insights regarding the impact of COVID-19 safety regulations on indoor air quality in a university classroom considering human patterns in view of developing proper mitigation strategies.

## 2. Materials and Methods

The experiments were performed at different times in the same classroom located at the 2nd floor of a fourth story building from the campus, which was finalized in 2015 with a modern building envelope respecting the standards. It is watertight and wind speed or temperature differences have less impact on the indoor measurements. The orientation of the classroom is south-east as seen on the topographic map (Appendix A). The classroom is equipped with a whiteboard with non-permanent markers, copy machine, video-projector, and three desktop computers that may contribute to the emissions and resuspension of PM (Appendix A). The dimensions of the classroom are typical for the campus in agreement with the regulations contained in the Romanian Civil Code. Consequently, the results can provide a general overview of the indoor conditions and a starting point for more detailed setups for assessing the health risks of students exposed to PM and other compounds. This will hopefully allow a better understanding of the factors and impacts related to indoor pollution during the educational process.

### 2.1. Monitoring Instrumentation

The PM measurements have been performed using a calibrated optical monitoring system and a particle counter. The TSI DustTrak^TM^ DRX 8533 EP monitor is an optical instrument that simultaneously measures in real-time the size segregated mass fraction concentrations i.e., PM1, PM2.5, PM4, PM10, and TPM over 0.001–150 mg/m^3^ as concentration range [25]. Particle count in the 0.3–20 μm range was achieved using an optical Lighthouse 3016 IAQ particle counter [26]. An HD32.3 portable datalogger from Delta OHM equipped with specific sensors for indoor microclimate analysis [27] was deployed in a corner opposite to the windows and accessing door in the same position as the PM monitors (Figure A1). 

Thermal microclimate was assessed by the following indices: WBGT Index, PMV Index (*Predicted Mean Vote*), and PPD (*Predicted Percentage of Dissatisfied*) provided in the DeltaLog 10 application (Figure A2 and Figure A3). The *Scharlau Index* was computed to estimate the thermal discomfort.

The monitoring of outdoor concentrations was performed continuously using optical monitors located at the campus of the university (Appendix A). Concomitantly, outdoor temperature, relative humidity, and wind characteristics were recorded using a Delta-T Devices weather station (Appendix A). Appendix A shows the time series of the recorded outdoor concentrations of PM2.5 and PM1 from which data were retrieved to compute average concentrations for the corresponding periods when the indoor measurements have been performed.

Furthermore, the reference indoor concentration of PM2.5 in the classroom without students and with windows and door closed was established for 3 h at approximately 11 μg m^−3^—coeff. of var. = 11.2% (e.g., Appendix A).

### 2.2. Physiological Characteristics of the Students and Classroom Description 

The interpretation of data required recording the classroom occupation (number of students, age and gender distribution, and physical activity levels of the occupants) during measurements (Table 1). In addition, all actions on ventilation and heating were recorded. Functional probes were retrieved using a questionnaire completed by each student with his consent to participate in the study. 

Indirect estimation of the physiological probes of students has been considered based on the respiratory characteristics such as minute ventilation (VE) and alveolar minute ventilation (AVE). Minute ventilation is the amount of air breathed per minute equaling approximately 6 L (normal rate of minute ventilation is 5 to 8 L/min). For example, tidal volumes of 500 to 600 mL at 12–14 breaths/min provide VE between 6.0 and 8.4 L. VE can double with light exercise, and it can exceed 40 L/min with heavy exercise. Around 2 L remain in the physiological dead space (VD) consisting of the upper airway and the mouth, and 4 L participate in gas exchange in the millions of alveoli constituting the alveolar ventilation [28]. Two respiratory rates have been taken into account considering a sedentary activity i.e., f_1_ = 10–11 breathes/min without a mask in 2019 and f_2_ = 12–14 breathes/min for the groups wearing a mask in 2021. Equations and used values are presented in Table 1. 

It was considered that air movement at the level of the classroom’s occupants must be at a temperature and velocity to ensure proper comfort. Likewise, the natural ventilation should be controllable to allow users to adjust the ventilation rate as required. Adjustments were achieved by the appropriate use of windows and a door. The classroom has a volume of 150 m^3^ and only natural ventilation was used by opening the windows at every 60 min for 5 min in the interval of three hours of practical works. The volume per person and the nominal occupancy are presented in Table 1.

### 2.3. Analysis, Modeling, and Statistics 

To obtain insights into the exchange rate of the air and other exogenous parameters that are influencing the indoor levels of PM2.5, a mass balance equation was considered using the parameters provided in [29], which monitored outdoor average concentrations of PM2.5 during indoor measurements and no filtration conditions: $$C_{i}=C_{o}\frac{P\cdot \lambda_{V}}{\lambda_{V}+\lambda_{D}+\lambda_{F}}+\frac{E}{(\lambda_{V}+\lambda_{D}+\lambda_{F})V}$$
where:
*C_i_*—concentration of PM_2.5_ (μg m^−3^);*C_o_*—ambient air concentration of PM_2.5_ (μg m^−3^) (according to the recorded average for each group: 69, 40, 25, and 32 μg m^−3^);*P*—penetration factor (unitless) = 0.97; *λ_V_*—infiltration ventilation rate (h^−1^) = 0.53;*λ_D_*—particle removal rate by deposition (h^−1^) = 0.39;*λ_F_*—particle removal rate by filtration (h^−1^) = 0—no filtration;*E*—PM_2.5_ total emissions from indoor sources (μg h^−1^) = 2.62;*V*—room volume (m^3^) = 150.


Variations of this model were successfully used to assess the potential inhaled dose rate of particles [30] including SARS-CoV-2 virus [31], and the parametric analysis to examine indoor PM2.5 concentrations according to flow rates and filter efficiency under various outdoor concentrations and indoor levels [32].

In the current work, ExpoFIRST version 2.0, which is a standalone tool available for download from the Exposure Factors module of the EPA-Expo-Box website (https://cfpub.epa.gov/ncea/risk/recordisplay.cfm?deid=322489, accessed on 5 June 2021), was considered for performing an exposure assessment by entering data in five tabs to estimate inhalation ADD (Figure A4). In Scenario Description tab, the route of exposure, dose metric, and exposure descriptor were described, and then, in tab 2: Media & Receptors, the inhalation rate type (e.g., long-term (daily), short-term (activity-specific); intensity level when short-term is selected), location/activity, and receptor characteristics (gender and age bins), respectively. The next tabs are tab 3: Contaminants for entering chemical-specific information, and tab 4: Exposure Factors that lists each receptor group based on age.

Descriptive statistics (average, coefficient of variation, skewness, and kurtosis) provided the main features of the dataset variability. Normal distribution was tested based on Sig. value results from two tests of normality (Kolmogorov–Smirnov and Shapiro–Wilk tests). A non-parametric Mann–Whitney U test was used to compare the differences between two independent groups of unequal size [33] to test whether two samples (without disposable masks and with masks) are likely to derive from the same population (H_0_: the populations are equal versus H_1_: the populations are not equal). The test ranks all of the dependent values and then uses the sum of the ranks for each group in the calculation of the test statistic. Linear trendlines of PM time series were compared based on R^2^. 

## 3. Results

The main results of this screening study were the characterization of PM load in the classroom together with the thermal comfort and the estimation of the ventilation parameters of the participating groups of students. These indicators will establish the framework for a modeling approach for simulating the exposure of students and academic personnel in the University’s classrooms.

### 3.1. Particulate Matter Load

The measurements performed in the classroom provided an overview of the PM load in the presence of various groups of students. Table 2 presents the statistics of the PM concentrations for various size fractions. PM1 ranged between 29 and 41 μg m^−3^ and PM10 between 30 and 42 μg m^−3^. It was observed that the particles belonged mostly to fine and submicrometric fractions, reaching a maximum value of 51 μg m^−3^ in Group D. While analyzing the corresponding time series, a decrease of the PM concentration towards the end of the lectures occurred both in A and B groups. PM levels remained constant in Group C and D in which students wore disposable face masks (Figure 1). 

The R^2^ of linear trendlines confirmed the decrease in A (R^2^ = 0.86) and B (R^2^ = 0.69) groups and the constancy in C (R^2^ = 0.00) and D (R^2^ = 0.06) groups. Table 3 shows the data collected by the particle counter, which correlates with the results regarding the mass concentrations provided by the PM optical monitor. Most of the particles corresponded to the 0.3, 0.5, and 1.0 micron classes. Groups C and D recorded preponderantly 0.3 micron particles (7,006,334 and 5,097,065), while A and B in 0.5 microns category (6,183,560 and 6,013,148), respectively. 

The mass balance approach provided approximations of the PM2.5 indoor concentration based on outdoor levels and other factors. Table 4 presents a comparison between modeled concentrations and the averaged values resulting from monitoring. While for Group A there is an agreement between both values and partially for Group B, the concentrations recorded during pandemics showed higher indoor concentrations suggesting that probably the rate of emissions from indoor sources could be higher. One potential cause could be the use of disinfectants and hand sanitizers, but this assumption must be checked through detailed experiments involving chemical speciation determination. 

### 3.2. Thermal Comfort 

The thermal comfort was estimated using a series of indices from which the most precise one reflects the influence of the physical and physiological variables on the thermal comfort i.e., PMV and PPD in relation to precise microclimatic conditions. Figure 2 shows the monitoring of various temperature types required for the computation of thermal comfort indices. 

Table 5 summarizes the results for each group showing a better thermal environment for groups C and D compared to the other two groups. PMV varied between 0.4 and 0.8, while PPD was between 9.2% and 16.4%. The A and B groups were characterized by acceptable thermal environment, while C and D reached the thermal well-being status.

### 3.3. Variability of Respiratory Characteristics between the Groups of Students

The results regarding the respiratory characteristics were retrieved indirectly based on the estimations using functional probes declared by the students. Table 6 shows the main statistics including the minute ventilation (VE) and alveolar minute ventilation (AVE) for each group. During three hours in the closed environment of the classroom, the alveolar minute ventilation of the groups including the participating students and the same lecturer were as follows Group A 11,884.3 L, Group B 11,123.9 L, Group C 8138.88 L, and Group D 8382.93 L. The latest two groups had higher ventilation rates due to the mask wearing. The means of alveolar minute ventilation rates ranged between 3.3 and 4.5 L/min with coefficients of variations between 13.3% and 24.5% depending on the structure of the group (age, sex, height, and weight). Mann–Whitney U tests showed significant differences (*p* < 0.05) between Group A and Group C (U-value = 34.5; U critical value of at *p* < 0.05 = 39; *p* = 0.025), respectively Group A and Group D (U-value = 36; U critical value of at *p* < 0.05 = 39; *p* = 0.030). The differences between B and C (U-value = 21; U critical value at *p* < 0.01 = 29; *p* = 0.003), and B and D (U-value = 12; U critical value at *p* < 0.01 = 29; *p* = 0.0005) groups had significant results at *p* < 0.01. 

The central tendency parameters such as age, body weight, and inhalation rates were used as inputs in the ExpoFIRST tool to assess ADR values for each group of students. Figure 3 shows the results for the age range and various age bins (20–30 years and 31–40 years). The average ADR for all groups was estimated as 6.58 × 10^−4^ mg/kg-day (CV = 14.3%) for 20–40 years range, while for 20–30 years bin the average was 7.38 × 10^−4^ mg/kg-day (CV = 18.9%) and 5.74 × 10^−4^ mg/kg-day (CV = 9.2%) for 31–40 years bin, respectively.

## 4. Discussion

In a room, the assessment of air exchange rate requires longer-term measurements to determine the prevailing conditions influenced by several exogenous parameters. The pollutants’ load in the air within a classroom has similarities with the outdoor air from surrounding sources coming in through infiltration and airing. An increase in ventilation rate facilitates the admission of outdoor pollutants and implicitly the removing and diluting effect of pollutants from indoor sources is counterbalanced by an increasing amount of pollutants originating from outdoor surroundings [34]. Indoor PM is considered to have preponderantly outdoor origins in schools [35]. In our study, the classroom is located in a building that is close to a major traffic road and some industrial sources [36]. However, due to the fact that the measurements were performed in a relatively colder period of time, the ventilation using windows was limited. Consequently, the density of human occupancy played an important role in the dynamics of indoor PM together with the inner sources. In the D group, which reached the highest PM concentration (41 μg m^−3^ PM1), the increase of PM concentrations could be attributed to the use of hand sanitizers and disinfecting solutions used to fulfill the anti-COVID epidemiological regulations (Figure 1d). The concentrations recorded during the lectures of the other three groups (29–36 μg m^−3^ PM1) may be considered as a benchmark for the classroom considered in this study regardless of the use of masks by the occupants. Other studies showed that the highest indoor PM10 levels were recorded in schools (33.0–97.2 μg m^−3^) compared to homes (10.8–37.7 μg m^−3^) and in schools, PM2.5 concentrations were considerably higher indoors (9.4–56.1 μg m^−3^) than outdoors (8.6–15.8 μg m^−3^) [37]. Our PM2.5 results (37, 30, 29, and 41 μg m^−3^ corresponding for each group) are consistent with the concentrations reported in the literature by [38], such as: indoor levels measured in Paris between 24.7 μg m^−3^ in homes and 34.5 μg m^−3^ in offices versus 24.3 μg m^−3^ and 28.3 μg m^−3^, respectively, for adults and children in Amsterdam, 21.6 μg m^−3^ in Boston, 24 μg m^−3^ in Zurich, and 21.9 and 36.7 μg m^−3^ in Grenoble (for summer and winter, respectively). The reports regarding the submicrometric particles still needs to be further addressed and this study can contribute to the completion of the indoor air quality characterization. In the current study, particle counting data correlated with the results regarding the mass concentrations and most of the particles belonged to the 0.3, 0.5, and 1.0 micron classes. Because the normal activity in a classroom during the practical works is frontal lesson with students sitting at the desks, it is a sedentary activity only interrupted by intervals and lesson changes can cause resuspension of the particles [24]. Even a reduced activity can have a remarkable impact on airborne particles with diameters higher than 5 μm [39]. Figure 1 shows that resuspension of these particles and coarse fraction were occurring more often in groups A and C, while the lowest was in Group D. The importance of occupants’ movement and of reduced ventilation in the indoor particulate level was underlined in [40]. On the other hand, indoor submicron particulate concentration can be rather correlated to the variability of the ventilation rate [24]. An increasing of air change per hour can involve an increment of indoor submicron particulate concentration especially in the moments when the external pollution reaches the top [34].

A decrease in the concentration of all the PM fractions was observed towards the end of the lectures, both in A and B groups with students that did not wear masks being before the COVID-19 pandemic. This could be attributed to the PM inhalation, filtering, and retaining capacity of respiratory mechanisms. PM concentration remained constant in groups C and D in which students wore disposable face masks. In [41], the protective effect of masks for everyday use made from different materials was tested against 20–1.000 nm particles with different velocities founding that 40–90% of aerosols were able to penetrate through these masks depending on the material and dampness. This might explain the massive presence of 0.3 micron particles during the lectures of groups C and D that wore masks. From a medical standpoint, there is a theoretical possibility of an airflow obstruction when wearing a mask [42]. Such effect should be further studied using proper instrumentation and protocols in conjunction with microclimate characterization [43]. Since the humans’ thermal sensation is connected to the thermal energy balance of the whole human body, such balance is related to physical activity and clothing. In this study, the feeling of heat in the body as a whole was predicted by calculating PMV and PPD. PPD predicted the percentage of people who would feel too hot or too cold in the classroom environment. PPD varied being higher in larger groups (*n* = 18) i.e., 15.7% and 16.4% and lower in smaller ones (*n* =10), 9.2% and 11.1%, respectively. Thermal comfort can influence the ventilation rates [44]. In this study, we estimated the ventilation rates using well-established equations that use the functional probes of the classroom’s occupants. Significant differences regarding the minute ventilation and alveolar minute ventilation were noticed between the groups that did not wear masks and the ones that did. 

High levels of PM have been recorded in four different periods with different groups of students and the submicrometric fraction was the most present. The groups of students that did not wear masks (before COVID-19 pandemic) were exposed to PM and there exists the possibility that the decrease of the PM concentrations could be related to the inhaling and retaining of the PM in the students’ lungs. The groups that wore masks have been better protected against PM and the PM remained constant during the 3 h of lecture taking into account the trendlines of the time series. In [45], the mean percent penetration of PM for each mask material ranged from 0.26% to 29%, depending on the flow rate and mask material. It is clear that the mask efficiency in PM filtering should be considered to avoid empirical assessments. 

The health effects of nanoparticles, which seem to be predominant in indoor environments, are better correlated to the surface area of the particles. More research should be performed involving personal monitoring of the respiratory functions of the participants by using spirometry and monitors that measure the dose of inhaled nanoparticles in the lungs (e.g., TSI NSAM 3550). Dynamic modeling should be also involved considering an indoor air mass balance with Monte Carlo simulations [29] involving also the complex effects of furniture/indoor thermal mass on building systems [46]. The goal is to achieve realistic models and aggregation methodologies for indoor mass elements [47].

On the other hand, another important issue in characterizing the indoor environments is the presence of mycotoxins that contributes to the ‘Sick Building Syndrome’ [48] caused by the biochemical manifestations of various micro-organisms [49].

There is a stringent need for low-cost portable tools for assessing the individual exposure to particulate matter (PM) and other air pollutants and mycotoxins of concerns in real-time [50]. The precision and reliability of the low-cost sensors for PM monitoring has increased in the last period having promising results [51,52].

Based on the study results, it is important to develop complex indoor surveys involving monitoring of air quality, thermal microclimate, and particularly the respiratory functions [53]. There are several useful insights that can be used in further experiments applying indoor air modeling [30,31], but more attention should be also given to mold and dampness, ventilation (determined from CO_2_ concentrations), and exposure to selected indoor air pollutants that are specific in the facilities of a university. 

Indoor air modeling supports the understanding of viruses’ inhalation transmission including SARS-CoV-2 in indoor environments by approximating how the inhaled amount of viruses is affected by factors such as room ventilation, breathing flow rate, gender, room size, aerosol size distribution, exposure time, level of exercise, and the type of vocal activity [31]. The lifelong sum of all the environmental contributions to human physiology and pathophysiology forms the exposome, which is a relatively new developed paradigm for studying the health consequences of the environment [32]. In addition to external environmental stressors, also lifestyle, socioeconomic status, and climate variations define the individual exposome [54]. In the current pandemic situation, there is also growing evidence that air pollution triggers comorbidities and increases the case fatality rate in patients with COVID-19 infection [55], whereas such epidemiological correlations are mostly missing for the characterization of various microenvironments including the indoor ones [56]. There is a need for developing multi-exposure concepts that include the most part of harmful environmental pollutants [54] besides PM. 

Minimizing indoor air pollutants is important for a productive learning environment in universities because of the potentially negative effects determined by VOCs, PM including allergens and molds, and combustion gases on the health and wellbeing of students [57]. Some of these pollutants are known for causing flu-like symptoms, headaches, nausea, and irritation of the eyes, nose, and throat along with their capacity to trigger asthma or allergy attacks [57]. Air pollutants have been found to negatively influence academic performance [58]. Moreover, the combined effects of pollutants on the risk of COVID-19 infection and the severity of respiratory or cardiovascular complications are not well understood so far. On this basis, the current study aimed to characterize the indoor microenvironment for a typical classroom in which students and academic personnel spend a lot of time during the educational process (approximately 728 h/academic year—2 semesters), both for PM levels and microclimate in the presence of various groups of students. The resulted data can be used to establish efficient ways to diminish the exposure of students to PM providing better conditions for learning. Knowing that the exposure to environmental risk factors, which is the external exposome, leads to changes of central biochemical pathways (stress response, oxidative stress, and inflammation) with associated health impact, it is important to have reliable instrumentation for estimating the exposure to harmful pollutants. Since classical health risk factors share similar pathomechanisms, students’ or professors’ pre-existing chronic diseases (e.g., diabetes or hypertension) may experience additive adverse health effects upon exposure to environmental stressors especially in indoor microenvironments with high levels of air pollutants [54]. It is expected that precise monitoring will provide supporting data for a reliable epidemiological assessment and this study employed robust monitors for evaluating the exposure to PM. Additionally, the microclimate can contribute to the aggravation of health effects. Consequently, maintaining adequate air exchange rates do not replace or reduce the need to control indoor sources of emission of harmful chemicals [15]. Ventilation and filtration control play certainly a critical role in the variability of indoor PM2.5 concentrations of indoor or outdoor origin in different residential environments including educational facilities [59]. In the current work, the experiments relied on ‘no filtration’ conditions because the building in which the room is located still does not have a ventilation/filtration unit. However, it is previewed that such a unit will be added in the near future providing safer levels of PM during the learning process. 

## 5. Conclusions

The instrumentation used in this study showed promising results for the characterization of the indoor microenvironments including university buildings. PM load should be estimated together with the thermal microclimate, and the physiological indicators, clothes, and metabolic characteristics of the classroom’s occupants. Such contextual data can provide useful insights on the indoor microclimate modifications, and give perspectives for the future of sensors that can support medical and occupational health and safety research in indoor environments.

We found that PM1 was the preponderant fraction ranging between 29 and 41 μg m^−3^ depending on the group of students in acceptable thermal environments according to the PPD and PMV indices. The particle counter recorded predominantly 0.3, 0.5, and 1.0 micron categories in the university classroom. 

COVID-19 impacts on indoor air quality were determined especially by the use of various disinfectant products in the presence of a lower number of students due to the epidemiological restrictions. Outdoor monitoring showed a clear reduction of exterior PM concentrations during March 2021 compared to the same month in 2019 also due to the imposed measures for mobility restraints to control the spread of the virus. 

The current approach has some limitations because individual testing of the students was not performed (spirometry, inhalation particularities, mask performance on filtering, and indoor microclimate influence) and some results have a broader context. Source apportionment, CO_2_ monitoring, and dynamic modeling should be also considered for a comprehensive characterization of the indoor exposure to PM. Due to epidemiological constraints, it was impossible to test the influence on mask wearing in the same group of students. 

Future work will take into account more experimental setups considering the aspects mentioned before. 

## Figures and Tables

**Figure 1 ijerph-18-07363-f001:**
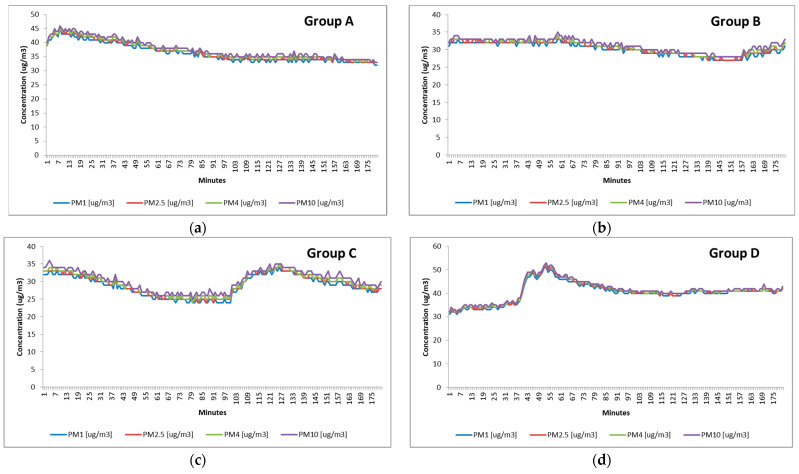
Time series of the concentrations of the size segregated mass fractions of particulate matter (μg m^−3^) recorded in a classroom using a TSI DustTrak^TM^ DRX 8533 EP during three hours of lectures and the equations of corresponding linear trendlines: (**a**) Group A: *y = −*0.0618*x +* 43.436 *(R*^2^ = 0.86*)*; (**b**) Group B: *y = −*0.0288*x +* 33.429 *(R*^2^
*=* 0.69*)* ; (**c**) Group C: *y = −*0.0002*x +* 30.3994 *(R*^2^
*=* 0.00*)*; (**d**) Group D: *y =* 0.0214*x +* 39.407 *(R*^2^
*=* 0.06).

**Figure 2 ijerph-18-07363-f002:**
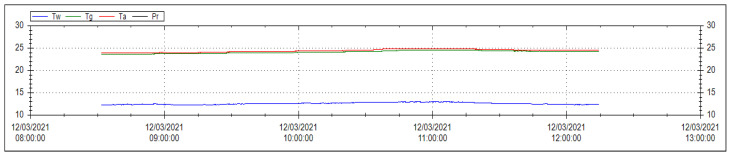
Example of the raw time series recorded by the DeltaOHM indoor microclimate system during the practical works of Group D (Tw—wet bulb temperature with natural ventilation; Tg—globe thermometer temperature; Ta—ambient temperature; Pr—vapor pressure).

**Figure 3 ijerph-18-07363-f003:**
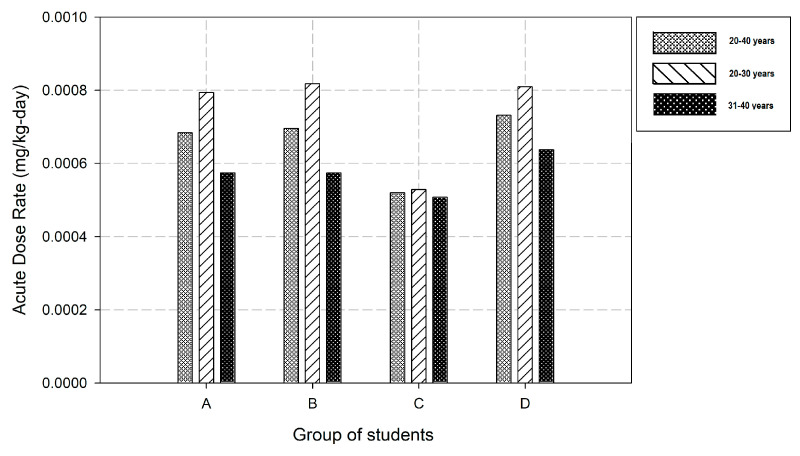
Acute dose rates estimated for the groups of students (A, B, D—mixed groups; C—only females); columns order: 20–40 years; 20–30 years; 31–40 years.

**Table 1 ijerph-18-07363-t001:** Setup of the experiment to determine the PM load and thermal microclimate in a university classroom in the presence of various groups of students without masks (2019) and with disposable face masks (2021).

Indicator	Descriptor
Location	Valahia University Campus, Targoviste, Romania—Classroom A227
Study period	3 h monitoring for each group in different days of March 2019 and in March 2021 (with anti-COVID protection measures)
Groups of students	2019—2 groups A (*n* = 17)—age (18–38); 13 females and 4 males. B (*n* = 17)—age (19–38); 14 females and 3 males. 2021—2 groups C (*n* = 9)—age (19–30); 9 females. D (*n* = 9)—age (19–43); 7 females and 2 males. Total = 52 students + 1 professor which was present in all groups (male)
Fields in the observation sheet completed by each student	age; weight; height; medication during the trial; presence of respiratory symptoms; physical effort before entering in the classroom; body temperature before entering the classroom.
Classroom	Surface (m^2^) = 50; Height (m) = 3; Volume (m^3^) = 150; Volume per person (m^3^/person) A-8.33 B-8.33 C-15 D-15 Nominal occupancy (person/m^2^) A-0.36 B-0.36 C-0.2 D-0.2
Respiratory characteristics	VE—minute ventilation (the total volume of gas entering or leaving the lungs per minute-L/min). VE = VT × f (where VT—tidal volume; f—respiratory rate). AVE—alveolar minute ventilation (L/min); AVE = VE − VD (where VD—physiological dead space); VD = 1 mL/pound IBW (where 1 pound = 0.45359237 kg; IBW—ideal body weight) IBW estimated using the formula of Devine (1974) https://www.calculator.net/ideal-weight-calculator.html, accessed on 5 June 2021 VT = 5–7 mL/kg f_1_ = 10–11 breathes/min considering a sedentary activity without mask f_2_ = 12–14 breathes/min considering a sedentary activity wearing mask

**Table 2 ijerph-18-07363-t002:** Concentration of the size segregated mass fractions of particulate matter (μg m^−3^) recorded in a classroom using a TSI DustTrak^TM^ DRX 8533 EP monitor during 3 h of lecture in the presence of students (Group A and B—without wearing masks; Group C and D—wearing disposal face masks).

PM	Group A	Group B	Group C	Group D
PM1 Average	36	30	29	41
*PM1 Minimum*	*32*	*27*	*24*	*31*
*PM1 Maximum*	*45*	*33*	*34*	*51*
*Coeff. of var.*	9.0	6.0	10.3	10.9
PM2.5 Average	37	30	29	41
*PM2.5 Minimum*	*33*	*27*	*24*	*32*
*PM2.5 Maximum*	*45*	*33*	*34*	*52*
*Coeff. of var.*	9.1	6.0	10.1	11.0
PM4 Average	37	31	30	41
*PM4 Minimum*	*33*	*28*	*24*	*32*
*PM4 Maximum*	*45*	*34*	*35*	*52*
*Coeff. of var.*	8.9	5.8	10.0	11.0
PM10 Average	38	31	30	42
*PM10 Minimum*	*33*	*28*	*25*	*32*
*PM10 Maximum*	*46*	*35*	*36*	*53*
*Coeff. of var.*	9.1	5.9	9.6	*10.8*

**Table 3 ijerph-18-07363-t003:** Particle counts and indoor microclimate recorded using the Lighthouse HH 3016 IAQ.

Size Fraction	Units	Group A	Group B	Group C	Group D
0.3 micron	(Counts)	3,453,673	3,584,492	7,006,334	5,097,065
0.5 micron	(Counts)	6,183,560	6,013,148	676,944	4,368,051
1.0 micron	(Counts)	349,388	371,017	240,909	416,673
2.5 micron	(Counts)	10,564	23,055	88,378	75,916
5.0 micron	(Counts)	2409	7736	30,054	38,044
10.0 micron	(Counts)	405	551	1280	4250
Sample Time	(s)	10,800	10,800	10,800	10,800
Sample Volume	(m^3^)	0.510	0.510	0.510	0.510
Temperature	(°C)	26.0	25.7	22.4	24.2
Relative Humidity	(%)	25.2	24.4	42.4	20.4

**Table 4 ijerph-18-07363-t004:** Results of the mass balance approach to estimate the indoor concentrations of PM2.5 (μg m^−3^).

Group	Monitored PM2.5 Outdoor Range	Average Outdoor PM2.5 Concentration (C_o_)	Indoor Modeled Concentration (C)	Indoor Measured Concentration	Required Outdoor Concentration to Reach Indoor Values
A	62–71	69	38.5	37	66
B	31–47	40	22.2	30	54
C	22–28	25	14.2	29	52
D	26–34	32	17.9	41	73

**Table 5 ijerph-18-07363-t005:** Thermal indicators estimated using the DeltaOHM HD32.3 during the practical works of students (3 h).

Indicator	Group A	Group B	Group C	Group D
PMV	0.8	0.7	0.5	0.4
PPD (%)	16.4	15.7	11.1	9.2
WBGT	22.4	20.3	16.5	16.1
Scharlau Index	13.9 °C	15.8 °C	6.5 °C	19.5 °C
Critical temperature	39.9 °C	41.8 °C	30.9 °C	43.7 °C

Thermal environment evaluation: PMV = +0.85; PPD =20%—acceptable thermal environment; −0.5 < PMV < +0.5 and PPD < 10%—thermal well-being.

**Table 6 ijerph-18-07363-t006:** Variability of functional probes and respiratory characteristics between the groups of students.

Students	Statistical Indicator	Age	Weight	Height	VT	VE	IBW	VD	AVE	Total AVE in 3 h
Units	-	years	kg	m	mL/kg	L/min	kg	L	L/min	L/min
Group A (*n* = 17)	Average	24.1	59.8	1.7	351.1	3.5	60	0.1	3.4	645.7
	Min.	18	35.4	1.5	212.4	2.1	41.5	0.1	2	382.3
	Max.	38	92	1.8	552	5.5	76	0.2	5.4	993.6
	Coeff. of var. (%)	29	24	5	24.5	24.5	15.2	15.1	24.5	24
	Skewness	1.3	0.6	−0.6	0.6	0.6	−0.3	−0.3	0.7	0.6
	Kurtosis	0.1	0.3	0.4	0.3	0.3	−0.3	−0.3	0.3	0.3
Group B (*n* = 17)	Average	22.8	58	1.7	345.4	3.4	59.7	0.1	3.3	597.9
	Min.	19	48	1.6	288	2.9	52.4	0.1	2.8	496.5
	Max.	38	80	1.8	480	4.8	75.1	0.2	4.7	839.6
	Coeff. of var. (%)	23.8	14.4	4.0	14.5	14.5	12.0	12.0	14.9	14.9
	Skewness	1.7	1.1	0.8	1.1	1.1	0.9	0.9	1.1	1.1
	Kurtosis	2.7	1.5	−0.5	1.5	1.5	−0.2	−0.2	1.7	1.7
Group C (*n* = 9)	Average	21.6	60.4	1.63	362.7	4.3	54.8	0.1	4.2	783.3
	Min.	19	49	1.5	294	3.5	45.1	0.1	3.4	635
	Max.	30	70	1.7	420	5	59.7	0.1	4.9	907.2
	Coeff. of var. (%)	19.8	13	2.9	13	13	7.8	7.8	13.3	13
	Skewness	1.6	−0.5	−1.6	−0.5	−0.5	−1.6	−1.6	−0.5	−0.5
	Kurtosis	1.0	−1.4	3.2	−1.4	−1.4	3.2	3.2	−1.4	−1.4
Group D (*n* = 9)	Average	24.8	64.7	1.7	388	4.7	62.5	0.1	4.5	813.3
	Min.	19	45	1.6	270	3.2	52.4	0.1	3.1	562.4
	Max.	43	80	1.9	480	5.8	75.2	0.2	5.6	1012.4
	Coeff. of var. (%)	40.6	21.7	5.1	21.7	21.7	12.8	12.8	22.1	22.1
	Skewness	1.6	−0.1	0.2	−0.1	−0.1	0.2	0.2	−0.1	−0.1
	Kurtosis	0.7	−2.0	−1.0	−2.0	−2.0	−1.0	−1.0	−2.0	−2.0

## Data Availability

The data presented in this study are available on request from the corresponding authors. The data are not publicly available due to the potential for further development of the methodology.

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
