# Peer review of "Performance Evaluation of Particulate Matter and Indoor Microclimate Monitors in University Classrooms under COVID-19 Restrictions"

_ijerph, 2021, doi:10.3390/ijerph18147363_

Round 1
Reviewer 1 Report
The authors have worked on clarifying the comments in an acceptable way. The article has gained in rigor and rotundity. Congratulations on the job.
Author Response
We would like to thank you for your support in improving our original manuscript. We felt that we should thank you in the Acknowledgments section for the useful comments and we did it. We hope that the revised version meets the scientific standards.
Reviewer 2 Report
1 Why do we need to evaluate the performance of particulate matter monitor in university classrooms under the limitation of COVID-19? What is the practical value and significance?
2 The study seems to have only evaluated the performance of different sensors. Further, how can the health risks of students exposed to PM be assessed?
3 Please highlight the contribution and value of this study compared with previous studies
4 Choose a classroom as an experimental case. Are the results representative?
Author Response
1 Why do we need to evaluate the performance of particulate matter monitor in university classrooms under the limitation of COVID-19? What is the practical value and significance?
A: Thank you for the evaluation of our manuscript and for the useful recommendations. We included more information in the discussion (modification with track changes):
The lifelong sum of all the environmental contributions to human physiology and pathophysiology forms the exposome, which is a relatively new developed paradigm for studying the health consequences of the environment [53]. In addition to external environmental stressors, also lifestyle, socioeconomic status, and climate variations define the individual exposome [54]. In the current pandemic situation, there is also growing evidence that air pollution triggers comorbidities and increases the case fatality rate in patients with COVID-19 infection [55], whereas such epidemiological correlations are mostly missing for the characterization of various microenvironments including the indoor ones [56]. There is a need for developing multi-exposure concepts that include the most part of harmful environmental pollutants [54] besides PM.
Minimizing indoor air pollutants is important for a productive learning environment in universities because of the potentially negative effects determined by VOCs, PM including allergens and molds, and combustion gases on the health and wellbeing of students [57]. Some of these pollutants are known for causing flu-like symptoms, headaches, nausea, and irritation of the eyes, nose, and throat along with their capacity to trigger asthma or allergy attacks [57]. Air pollutants have been found to negatively influence academic performance [58]. Also, the combined effects of pollutants on the risk of COVID-19 infection and the severity of respiratory or cardiovascular complications are not well understood so far. On this basis, the current study aimed to characterize the indoor microenvironment for a typical classroom in which students and academic personnel spend a lot of time during the educational process (approximately 728 hours/academic year – 2 semesters), both for PM levels and microclimate in the presence of various groups of students. The resulted data can be used to establish efficient ways to diminish the exposure of students to PM providing better conditions for learning.
2 The study seems to have only evaluated the performance of different sensors. Further, how can the health risks of students exposed to PM be assessed?
A: We included the following text in the discussion:
Knowing that the exposure to environmental risk factors, which is the external exposome, leads to changes of central biochemical pathways (stress response, oxidative stress, inflammation) with associated health impact, it is important to have reliable instrumentation for estimating the exposure to harmful pollutants. Since classical health risk factors share similar pathomechanisms, students’ or professors’ pre-existing chronic diseases (e.g. diabetes or hypertension) may experience additive adverse health effects upon exposure to environmental stressors especially in indoor microenvironments with high levels of air pollutants [54]. Additionally, the microclimate can contribute to the aggravation of health effects. It is expected that precise monitoring will provide supporting data for a reliable epidemiological assessment and this study employed robust monitors for evaluating the exposure to PM.
3 Please highlight the contribution and value of this study compared with previous studies
Following your recommendation, we have extended the discussion including 6 relevant articles from which 2 have been published in IJERPH:
- A: Wild, C.P. Complementing the genome with an "exposome": the outstanding challenge of environmental exposure measurement in molecular epidemiology. Cancer Epidemiol Biomarkers 2005 14(8), 1847-50. doi: 10.1158/1055-9965.EPI-05-0456.
- Daiber, A; Kuntic, M; Hahad, O; Delogu, L.G.; Rohrbach, S.; Di Lisa, F.; Schulz, R.; Münzel, T. Effects of air pollution particles (ultrafine and fine particulate matter) on mitochondrial function and oxidative stress - Implications for cardiovascular and neurodegenerative diseases. Arch Biochem Biophys. 2020 15; 696, 108662. doi: 10.1016/j.abb.2020.108662.
- Travaglio, M.; Yu, Y.; Popovic, R.; Selley, L.; Leal, N.S.; Martins, L.M. Links between air pollution and COVID-19 in England. Environ Pollut. 2021, 268(Pt A), 115859. doi: 10.1016/j.envpol.2020.115859.
- Yao, Y.; Chen, X.; Chen, W.; Wang, Q.; Fan, Y.; Han, Y.; Wang, T.; Wang, J.; Qiu, X.; Zheng, M.; Que, C.; Zhu, T. Susceptibility of individuals with chronic obstructive pulmonary disease to respiratory inflammation associated with short-term exposure to ambient air pollution: A panel study in Beijing. Sci Total Environ. 2021 20, 766, 142639. doi: 10.1016/j.scitotenv.2020.142639.
- Cincinelli, A.; Martellini, T. Indoor Air Quality and Health. J. Environ. Res. Public Health 2017, 14, 1286. https://doi.org/10.3390/ijerph14111286
- Mullen, C.; Grineski, S.E.; Collins, T.W.; Mendoza, D.L. Effects of PM2.5 on Third Grade Students’ Proficiency in Math and English Language Arts. Int. J. Environ. Res. Public Health 2020, 17, 6931. https://doi.org/10.3390/ijerph17186931
We hope that this completion is meeting your expectations.
4 Choose a classroom as an experimental case. Are the results representative?
A: Thank you for mentioning. We included a paragraph in the methodology pointing out the necessity of establishing a framework for further detailed setups to allow the development of a model with increased generalization capabilities. We used a typical classroom that is mostly common in the campus, the potential differences being the exposition and the floor.
“The dimensions of the classroom are typical for the campus in agreement to the regulations contained in the Romanian Civil Code. Consequently, the results can provide a general overview of the indoor conditions and a starting point for more detailed setups for assessing the health risks of students exposed to PM and other compounds. This will hopefully allow a better understanding of the factors and impacts related to indoor pollution during the educational processes.”
We are thanking you for the very useful comments. We thanked your contribution in the acknowledgment section.